# DNA Methylation and Prospects for Predicting the Therapeutic Effect of Neoadjuvant Chemotherapy for Triple-Negative and Luminal B Breast Cancer

**DOI:** 10.3390/cancers15051630

**Published:** 2023-03-06

**Authors:** Vladimir O. Sigin, Alexey I. Kalinkin, Alexandra F. Nikolaeva, Ekaterina O. Ignatova, Ekaterina B. Kuznetsova, Galina G. Chesnokova, Nikolai V. Litviakov, Matvey M. Tsyganov, Marina K. Ibragimova, Ilya I. Vinogradov, Maxim I. Vinogradov, Igor Y. Vinogradov, Dmitry V. Zaletaev, Marina V. Nemtsova, Sergey I. Kutsev, Alexander S. Tanas, Vladimir V. Strelnikov

**Affiliations:** 1Research Centre for Medical Genetics, 115522 Moscow, Russia; 2N. N. Blokhin National Medical Research Center of Oncology, 115478 Moscow, Russia; 3Laboratory of Medical Genetics, I. M. Sechenov First Moscow State Medical University (Sechenov University), 119992 Moscow, Russia; 4Cancer Research Institute, Tomsk National Research Medical Center, Russian Academy of Sciences, 634009 Tomsk, Russia; 5Regional Clinical Oncology Dispensary, 390011 Ryazan, Russia; 6Department of Pathological Anatomy, Ryazan State Medical University, 390026 Ryazan, Russia

**Keywords:** breast cancer, DNA methylation, RRBS, MSRE-qPCR, predictive markers, neoadjuvant chemotherapy

## Abstract

**Simple Summary:**

Breast cancer (BC) is a group of diseases heterogeneous in morphology, progression, survival, and response to therapy. Although BC is among the most exhaustively studied cancers, there is still a lack of molecular markers to predict its response to neoadjuvant chemotherapy (NACT). Tumor development is determined by alterations not only of its genome, but of its epigenome as well. In order to identify epigenomic markers of BC NACT effectiveness, we have applied genome-wide DNA methylation screening of tumors in cohorts of NACT responders and nonresponders. Combining several of the most informative DNA methylation markers, we suggest tiny diagnostic panels that may be readily implemented in diagnostic laboratories. We also demonstrate that clinical characteristics predictive of NACT response, such as the clinical stage and lymph node status, are independently additive to the epigenetic classifiers and in combination improve prediction.

**Abstract:**

Despite advances in the diagnosis and treatment of breast cancer (BC), the main cause of deaths is resistance to existing therapies. An approach to improve the effectiveness of therapy in patients with aggressive BC subtypes is neoadjuvant chemotherapy (NACT). Yet, the response to NACT for aggressive subtypes is less than 65% according to large clinical trials. An obvious fact is the lack of biomarkers predicting the therapeutic effect of NACT. In a search for epigenetic markers, we performed genome-wide differential methylation screening by XmaI-RRBS in cohorts of NACT responders and nonresponders, for triple-negative (TN) and luminal B tumors. The predictive potential of the most discriminative loci was further assessed in independent cohorts by methylation-sensitive restriction enzyme quantitative PCR (MSRE-qPCR), a promising method for the implementation of DNA methylation markers in diagnostic laboratories. The selected most informative individual markers were combined into panels demonstrating cvAUC = 0.83 (*TMEM132D* and *MYO15B* markers panel) for TN tumors and cvAUC = 0.76 (*TTC34*, *LTBR* and *CLEC14A*) for luminal B tumors. The combination of methylation markers with clinical features that correlate with NACT effect (clinical stage for TN and lymph node status for luminal B tumors) produces better classifiers, with cvAUC = 0.87 for TN tumors and cvAUC = 0.83 for luminal B tumors. Thus, clinical characteristics predictive of NACT response are independently additive to the epigenetic classifier and in combination improve prediction.

## 1. Introduction

According to GLOBOCAN statistics (185 countries), 19.3 million new cases of cancer were diagnosed in 2020, of which 11.7% (2,261,419 of cases) were breast cancer (BC), and the death rate from BC was 684,996 people per year, which is more than in previous years. In women, BC occupies a leading position in both new cancer cases and fatalities [1].

Despite advances in diagnosis, surgical treatment, and systemic therapy, the main cause of death is resistance to existing therapies [2].

An approach to improve the effectiveness of therapy in patients with aggressive subtypes of breast cancer, such as luminal B or triple-negative BC (TNBC), is the administration of neoadjuvant (preoperative) chemotherapy (NACT). According to large clinical studies, when a pathomorphological complete response (pCR) is achieved because of NACT, the survival of patients with aggressive subtypes of breast cancer is close to the survival of patients with more favorable subtypes, compared with patients in this group with a residual tumor [3]. In addition, in patients with locally advanced breast cancer, NACT is a mandatory component of treatment of all breast cancer subtypes; it allows for a reduction in the volume of the primary tumor and increases the frequency of organ-preserving operations. The response to neoadjuvant chemotherapy for aggressive subtypes is less than 65% according to large breast cancer clinical trials [4].

At the same time, it has been shown that DNA methylation is an early and frequent event in carcinogenesis [5].

DNA methylation (5-methylcytosine, 5-mC) is one of the best-studied epigenetic modifications that acts directly on genomic DNA, where a CH_3_- group is added at the C5 position of the cytosine ring in the palindromic CpG dinucleotide. Methylation is known to affect gene expression through the regulation of gene transcription [6]. Back in 2003, Peter Laird described how recent advances in understanding the role of methylation in cancer could one day lead to many powerful biomarkers based on DNA methylation, especially for use as diagnostic markers in oncology [7]. This belief was based on a number of characteristics of aberrant DNA methylation that make it a promising source of biomarkers: abnormal DNA methylation is an early and frequent event in carcinogenesis, is easy to detect using well-established methods, and is stable in fixed samples over time, and the DNA molecule is stable double-stranded nucleic acid and can be detected in various body fluids [5].

Although luminal B tumors are highly proliferative, they are less likely to respond to NACT, as treatment resistance is common in this subtype. Therefore, luminal B is one of the breast cancer subtypes that needs new markers to personalize the prescription of preoperative chemotherapy and identify those patients who can get the maximum benefit from treatment with a minimum effect of chemotherapy drug toxicity.

Neoadjuvant chemotherapy remains the gold standard of care for patients with TNBC, but is characterized by limited efficacy, a narrow response time, and significantly toxic profiles. TNBC is the most aggressive subtype with a higher metastasis rate, early recurrence, and poor overall survival, accounting for about 15–20% of all breast cancer cases. Unfortunately, only one in three patients respond successfully to treatment [8], which makes it urgent to find alternative methods for predicting the response of each patient in order to provide patients with more individualized medical care. The task of forming epigenetic diagnostic panels for predicting the effectiveness of NACT in breast cancer patients can be effectively solved using a modern method of genome-wide analysis of DNA methylation, reduced representation bisulfite sequencing (RRBS). RRBS increases the relative informational value of DNA methylation analysis compared to whole-genome bisulfite sequencing (WGBS): each RRBS sequence read includes at least one informative CpG position. Theoretically, RRBS is better than WGBS and is applicable to large-scale studies of DNA methylation and the search for markers of epigenetic processes in health and disease, since RRBS focuses on functionally significant methylation in CpG islands, ignoring less clinically significant regions [9].

## 2. Materials and Methods

### 2.1. Patients Treatment and Tumor Samples

This study included 156 patients with early (T_1-2_N_0-1_M_0_) and locally advanced (T_2-4_N_2-3_M_0_) TNBC or with luminal B breast cancer of the IIA-IIIC clinical stages. Genome-wide bisulfite sequencing was performed on 73 breast cancer biopsy specimens (discovery cohort) obtained before neoadjuvant chemotherapy to select genome regions for the further formation of DNA methylation panels of a limited number of the most informative markers. On an independent cohort of 83 core biopsy specimens taken before NACT, we performed an evaluation of the diagnostic significance of DNA methylation panels by quantitative multilocus methylation-sensitive restriction enzyme quantitative PCR (MSRE-qPCR).

The discovery cohort consisted of 29 TNBC samples and 44 samples of luminal B immunohistochemical subtypes (Table 1). The NACT scheme for a group of patients with TNBC was eight courses of doxorubicin, cisplatin, and paclitaxel. Patients with luminal B subtype tumors received 2 to 4 courses of NACT with CAX (cyclophosphamide + doxorubicin + xeloda) or FAC (fluorouracil + doxorubicin + cyclophosphamide).

The independent cohort consisted of 48 TNBC samples and 35 samples of the luminal B subtype (Table 1). In this cohort, the NACT regimen for TNBC patients was 8 courses of doxorubicin, cisplatin, and paclitaxel. Patients with luminal B tumors received 4–6 courses of adriamycin (doxorubicin) and cyclophosphamide.

A statistical comparison of clinical features between the discovery and independent cohorts determined significant differences in patient’s ages (*p* = 0.02) and NACT response (*p* = 0.03) for TN breast cancer and luminal B BC (*p* < 0.001). No other statistical differences were found.

### 2.2. Evaluation of Neoadjuvant Chemotherapy

The effect of neoadjuvant chemotherapy was assessed based on the results of clinical examination, ultrasound, and mammography. For luminal B tumors of a discovery cohort, the response to NACT was classified according to the RECIST criteria for evaluating the response in solid tumors (version 1.1) [10]. A complete regression was defined as the complete disappearance of the primary tumor and lymph node metastases. Such tumors were classified as sensitive to NACT. A partial response was defined as tumor reduction by ≥30% and such tumors were also referred to as sensitive. NACT-resistant tumors included those with post-NACT stabilization (<30% tumor reduction or <20% tumor size increase) and progression (≥20% tumor size increase). For TNBC, the evaluation was performed according to the Lavnikova scale. Samples with a pathological complete response (pCR) were assigned to the group of tumors sensitive to NACT and all the rest to the group of resistant tumors. The response to NACT of tumors from an independent cohort of both triple-negative and luminal B subtypes was assessed using the Residual Cancer Burden (RCB) scale [11]. The sensitive group included samples with RCB0-RCB2; RCB3 tumors were classified as resistant.

### 2.3. DNA Isolation

DNA was isolated by the classical phenol–chloroform method. After the complete dissolution of the precipitate, DNA concentration was measured on a Qubit 4 fluorimeter (Thermo Fisher Scientific, Waltham, MA, USA) using Qubit DNA BR Assay Kits (Thermo Fisher Scientific, Waltham, MA, USA).

### 2.4. Genome-Wide DNA Methylation Analysis

Genome-wide bisulfite DNA sequencing was performed according to the previously described XmaI-RRBS technology [12] on an Ion Torrent PGM sequencer (Thermo Fisher Scientific, USA).

Briefly, the DNA was treated with restriction endonuclease XmaI, and then the sticky ends were partially blunted with methylated cytosines using a 3′-5′ Klenow exo- and ligated with adapters containing methylated cytosines (presented in Appendix A). The libraries obtained were selected by length to obtain a fraction of fragments with an insert size of 100–200 bp, with subsequent bisulfite conversion using the Qiagen EpiTect Bisulfite Kit (Qiagen, Hilden, Germany). To avoid the nonspecific priming of the 3′ ends of DNA fragments in the polymerase reaction to follow, those were blocked with chain-terminating dideoxynucleotides using the SNaPshot Multiplex Kit (Thermo Fisher Scientific, USA). Then, RNase A (Sigma-Aldrich, St. Louis, MO, USA) and alkaline phosphatase (SibEnzyme, Novosibirsk, Russia) were used to remove the carrier RNA used in the EpiTect Bisulfite Kit protocol and dephosphorylate residual ddNTPs, respectively. The final libraries were amplified by PCR, the number of cycles of which was determined based on preliminary measurements by quantitative PCR. The resulting libraries were quantified on a Qubit 4.0 fluorimeter (Thermo Fisher Scientific, USA).

Emulsion PCR was performed on an Ion OneTouch 2 instrument using Ion OneTouch 200 kits (Thermo Fisher Scientific, USA) according to the manufacturer’s instructions. The resulting Ion Sphere Particles (ISPs) were enriched at 37 °C using an Ion OneTouch ES system (Thermo Fisher Scientific, USA).

The sequencing results were processed using standard Ion Torrent Suite software. Bismark software [13], which was used to align the obtained reads on the GRCh37/hg19 human genome sequence using the Bowtie 2 aligner [14].

### 2.5. DNA Methylation Markers’ Selection Criteria

From the RRBS results, CpG dinucleotides were selected, for which no more than 20% of missing data on the methylation level were observed in the entire set of studied samples. Hierarchical clustering was performed for thus obtained CpG dinucleotides using the normalized pairwise Manhattan distance and the ward D2 agglomerative method [15]. To form a list of candidate markers, we selected loci whose differential methylation significantly differed (nominal *p* < 0.05) in groups of tumors resistant and sensitive to neoadjuvant chemotherapy after applying the Mann–Whitney test and for which there was information on the methylation status of 4 or more consecutive CpG pairs in the region. The difference in *TERT* and *DPYS* methylation levels between the groups did not pass the threshold of statistical significance; however, these loci were included in the further study due to their biological role.

### 2.6. Design of Primers for PCR Multiplex Assays

To select genome regions for inclusion in MSRE-qPCR-based assays, the following criteria were used: location in the region of the XmaI-RRBS library fragment (between two XmaI restriction enzyme sites), the presence of at least 3 recognition sites for methylation-sensitive restriction enzyme BstHHI; length of potential PCR product of no more than 400 bp; and flanking areas of a target locus not containing BstHHI recognition sites of at least 50 bp.

Alongside target loci (DNA methylation markers), each multiplex MSRE-qPCR assay was designed to include a positive internal control (PC) and a digestion efficacy control (DC). Loci for the PC were selected from 4 regions of the genome not containing recognition sites for the restriction enzyme used (regions of the CpG islands of the *SBNO2*, *LINC00493*, and *CLDND1* genes and the intergenic CpG-island on chromosome 18q21.33). The DCs were selected from 5 genome regions, each containing 2 restriction enzyme recognition sites and nonmethylated both in normal tissues and in tumors (CpG islands of the *TMEM158*, *ANO10*, and *ABHD5* genes) (Appendix A).

DC scores were not used for target methylation level calculation. They were only used for DNA hydrolysis quality control. Samples that demonstrated ΔCt < 6 for the DCs in digested versus nondigested aliquots were excluded from the final analysis. Primers and TaqMan probes were designed using MPprimer 1.4 software [16]. All candidate markers were combined into 11 pools according to the compatibility matrix in multilocus PCR. Each pool contained 1 to 3 target loci and 2 internal controls, a PC and DC (Appendix A).

### 2.7. Processing of DNA with Methylation-Sensitive Restriction Enzyme and MSRE-qPCR

As a template for qPCR, we used DNA samples hydrolyzed with BstHHI (SibEnzyme, Russia) restriction enzyme (GCG/C recognition site), as well as intact DNA, without the addition of BstHHI but maintaining the reaction conditions and the composition of the reaction mixture. Enzymatic hydrolysis was carried out in a total volume of 35 μL, containing 10 units of restriction endonuclease, 3.5 μL of 10× SE-buffer Y (SibEnzyme, Russia), and 60 ng of DNA for 12 h at 50 °C.

PCR was performed on a QuantStudio 5 (Thermo Fisher Scientific, USA) in GenPak PCR Core kit (Isogen, Moscow, Russia) plates, in 20 µL reaction volumes containing 10 µL of PCR Diluent (supplied with GenPak PCR Core kit), 300 nM of each primer, 200 nM of each TaqMan probe, and 10 ng of the input DNA template. The PCR program for all pools was unified to simplify the use of the laboratory protocol: the reaction was heated at 95 °C for 5 min and 50 PCR cycles were performed as follows: primary template denaturation at 95 °C for 30 s and annealing combined with elongation for 2 min with signal detection at this stage. Primer annealing temperatures were selected empirically for each pool.

The determination of the methylation level according to MSRE-qPCR results was performed using the ΔΔCy0 method. The methylation level of the target locus (tgt) relative to the PC locus, which does not contain restriction enzyme recognition sites, was determined by the following formulas in pairs of samples, MS (treated with methylation-sensitive restriction enzyme) and mock (without enzyme treatment):$$\Delta Cy0_{PC}=Cy0_{PC}^{MS}-Cy0_{PC}^{mock}$$
where $Cy0_{PC}^{MS}$ is the *Cy*0 value for the PC locus when treated with a methylation-sensitive restriction enzyme, and $Cy0_{PC}^{mock}$ is the *Cy*0 value for the PC locus without treatment with a methyl-sensitive restriction enzyme;
$$\Delta Cy0_{tgt}=Cy0_{tgt}^{MS}-Cy0_{tgt}^{mock}$$
where $Cy0_{tgt}^{MS}$ is the *Cy*0 value for the target locus when treated with a methylation-sensitive restriction enzyme, and $Cy0_{tgt}^{mock}$ is the *Cy*0 value for the target locus without treatment with a methyl-sensitive restriction enzyme;
$$\Delta \Delta Cy0_{tgt}=\Delta Cy0_{tgt}-\Delta Cy0_{PC}\;B_{tgt}=E^{-\Delta \Delta Cy0_{tgt}};\;$$
where the $B_{tgt}$ is the methylation level of the target locus and E is the PCR efficiency.

### 2.8. DNA Methylation Marker Panels’ Development Criteria

To combine individual DNA methylation markers into panels, markers were selected that demonstrated cvAUC > 0.5 according to the results of MSRE-qPCR. From the resulting panels, the best ones were selected in terms of cvAUC.

### 2.9. Statistical Analysis

Statistical data processing and visualization of the results were carried out using the programming language for statistical data processing, R.

Clinical features between the discovery and independent cohorts were compared using the Wilcoxon test for patient’s age and Chi-squared test for other features.

Over-representation analysis (ORA) was carried out using g:Profiler portal (https://biit.cs.ut.ee/gprofiler/gost (accessed on 23 February 2023)).

DNA methylation markers were selected from XmaI-RRBS data using the Wilcoxon–Mann–Whitney test.

Point-biserial correlation was used to test the independence between DNA methylation markers and the clinical parameters of patients. Differences were considered significant at *p* < 0.05. To determine the correlations between the methylation level of epigenetic markers and clinical/morphological features, the coefficient ρ was used.

Higher ρ coefficients denote a stronger magnitude of relationship between variables. Smaller ρ coefficients denote weaker relationships. Positive correlations denote a relationship that travels at the same trajectory. As one value goes up, the other value goes up.

When assessing the strength of correlations, the Chaddock scale was used. An FDR correction for multiple hypothesis testing was applied.

A Chi-square test was used to assess the significance of the association of clinical features and the response to NACT. Differences were considered significant at *p* < 0.05. To determine the relationship between the response to therapy and clinical/morphological features, the coefficient φ was used.

The φ coefficient is similar to the correlation coefficient in its interpretation. The φ coefficient value can be between 0 and 1. A coefficient of zero (0) indicates that the variables are perfectly independent. The larger the coefficient, the closer the variables are to forming a pattern that is perfectly dependent, which is 1.

When assessing the strength of the relationship of the correlation coefficient, the Chaddock scale was used.

As a classification algorithm, logistic regression with a standard threshold of 0.5 was used to determine if a sample belongs to the group of sensitive or resistant to NACT.

For assessing goodness-of-fit epigenetic and combined marker panels logistic regression models, a likelihood ratio test was used.

Classification quality was analyzed using ROC curves (ROC analysis). Due to the moderate number of patients in the cohorts tested in our study, cross-validation was used to characterize individual markers and their combinations. The procedure of the cross-validation of models was carried out by the 100-fold randomized division of an independent cohort of breast cancer samples into training and a test in the ratio of 75% and 25%, respectively. A cross-validated AUC (cvAUC) R package was used to calculate ROCcurves (https://github.com/ledell/cvAUC (accessed on 15 October 2022)). The most favorable sensitivity and specificity points for classifiers were obtained using the Youden index.

The NACT response score (*NRS*) was determined by the following formula:$$NRS=\frac{\exp\left(PRS\right)}{1+\exp\;\left(PRS\right)}$$
where *PRS* is a patient’s risk score:$$PRS=\beta_{0}+\sum_{i=1}^{n}(\beta_{marker_{i}}*\;marker_{i})$$
where β_0_ is an intercept of logistic regression model, β*_marker_* is a regression coefficient, *marker* is a DNA methylation marker’s methylation value measured by MSRE-qPCR, and *n* is a number of elements in the diagnostic panels.

## 3. Results

### 3.1. An Unbiased Genome-WIDE DNA Methylation Markers Screening by XmaI-RRBS

In this study, genome-wide bisulfite sequencing was performed on 73 breast cancer biopsy specimens obtained prior to neoadjuvant chemotherapy. The samples constituted a discovery cohort for a genome-wide NACT response markers search from 29 samples of triple-negative breast cancer and 44 samples of luminal B subtypes. Based on the results of XmaI-RRBS sequencing, we obtained data on average of 118 million base pairs in 750,000 reads per sample with an average depth of 50.

The sequencing data obtained were placed in the GEO database, under the numbers GSE123712 and GSE123828, and can be used by other research teams to conduct comparative studies and explore breast cancer epigenetics.

Upon the results of XmaI-RRBS, 290 differentially methylated genes marking NACT-resistant and -sensitive triple-negative (Figure 1) tumors and 202 genes marking luminal B (Figure 2) tumors were identified in the discovery cohort. Gene lists are presented in the Appendix A.

Candidate markers enrichment for specific biological processes, molecular functions, and KEGG pathways was assessed using an over-representation analysis (Appendix A).

Based on the presented lists of differentially methylated genes and taking into account the criteria for selecting genome regions for inclusion in panels, the set of candidate DNA methylation markers was determined to further design panels of a limited number of markers of TNBC sensitivity to NACT: *CDO1*, *CLEC14A*, *DLEU2*, *BNC1*, *PRKCB*, *GMDS*, *TERT*, *TTC34*, *TMEM132D*, *VGLL4*, *ABCA3*, *DPYS*, *IRF4*, *TMEM132C*, *SFRP2*, *SOX21*, and *MYO15B* (17 markers) and for luminal B: *LTBR*, *NRN1*, *TERT*, *TTC34*, *TMEM132D*, *VGLL4*, *ABCA3*, *DPYS*, *IRF4*, *TMEM132C*, *SFRP2*, and *MYO15B* (12 markers).

In the generated list, 11 markers (*TERT*, *TTC34*, *TMEM132D*, *VGLL4*, *ABCA3*, *DPYS*, *IRF4*, *TMEM132C*, *SFRP2*, *SOX21*, and *MYO15B*) potentially mark sensitivity to NACT of tumors of both BC subtypes under study.

The predictive value of the above candidate differential methylation markers was further assessed by MSRE-qPCR, as far as we deem it an optimal method for the practical implementation of differential DNA methylation assays.

### 3.2. Predictive Value of Differential Methylation Markers Assessed by MSRE-qPCR

Considering all the requirements described in the Materials and Methods section, 11 multiplex MSRE-qPCR assays (pools of primers/probes) were designed (G1–G5, TN1–TN4, TN6, and LB1). The G (general) pools include markers that discriminate both TN and luminal B tumors in terms of response to NACT; the TN (triple-negative) pools include markers exclusive for TNBC and the LB (luminal B) pools for the luminal B subtype. Primers and probes for each of the pools are presented in Appendix A, and related functions of the marker genes are listed in Table 2.

Using the MSRE-qPCR method, we measured the methylation level of individual candidate markers (for vast differentially methylated regions, several loci were assessed, within the same gene) of tumor sensitivity to neoadjuvant chemotherapy in an independent cohort (*n* = 83) of TN and luminal B tumors. The predictive value of individual markers was characterized in terms of cvAUC (cross-validated area under the ROC curve), sensitivity, and specificity. The results in TN and luminal B tumor groups are shown in Appendix A, respectively.

The highest predictive values in terms of predicting the sensitivity to NACT of triple-negative tumors, were shown by methylation markers of the *TMEM132D*, *ABCA3*, *DPYS*, *MYO15B*, *GMDS*, *CDO1*, *SFRP2*, *DLEU2*, *IRF4*, *TMEM132C*, and *VGLL4* genes (cvAUC values are in order decreasing from 0.72 to 0.59). All markers demonstrate hypomethylated status in the group of NACT-sensitive tumors compared to the group of resistant ones. For the luminal B group of tumors, most informative were the methylation markers of the *LTBR*, *VGLL4*, *DPYS*, *CLEC14A*, and *TTC34* genes (cvAUC in descending order from 0.69 to 0.56). The markers of *LTBR*, *VGLL4*, and *CLEC14A* showed a hypomethylated status in the group of sensitive tumors compared with the resistant group, while the *DPYS* marker showed a hypermethylated status in sensitive tumors.

### 3.3. Development of DNA Methylation Marker Panels

The accuracy of diagnostics can be significantly improved by combining several markers into panels [17]. To achieve this, we combined the most informative individual DNA methylation markers in panels (Appendix A). When forming combinations of markers, no more than four markers per panel were taken, as far as the qPCR format usually allows no more than six detection channels (two channels are occupied by PC and DC controls in MSRE-qPCR assays).

Combining individual markers into panels improved the quality of NACT response classifiers for triple-negative tumors up to cvAUC = 0.83 (*TMEM132D* and *MYO15B* markers panel, with sensitivity and specificity both equal 0.76) and up to cvAUC = 0.76 (*TTC34*, *LTBR*, and *CLEC14A* markers panel, with sensitivity of 0.7 and specificity of 0.79) for luminal B tumors.

The same classifiers validated using the RRBS results demonstrated similar performance: cvAUC = 0.83 with sensitivity of 0.87 and specificity of 0.67 for TMEM132D and MYO15B; cvAUC = 0.67 with sensitivity of 0.6 and specificity 0.75 for TTC34, LTBR, CLEC14A (Appendix A).

### 3.4. Combination of Epigenetic, Clinical, and Morphological Markers Improves Prediction of NACT Response in BC

Statistical analysis has revealed no significant associations between the level of methylation of the studied markers and clinical and morphological characteristics of breast cancer, such as the patient’s age, tumor size (T), regional lymph nodes status (N), or stage, for both TN and luminal B tumor groups, after applying multiple-testing correction (Appendix A). This observation allows us to suggest that clinical and morphological characteristics predictive of the NACT response, if any, would be independently additive to the epigenetic classifier, and combining epigenetic, clinical, and morphological markers would further improve prediction of NACT response.

Correlations between different clinical features and the response to NACT in our cohorts are presented in Figure 3 and Figure 4. In the TN subtype, a significant correlation (*p* < 0.05) was found between the tumor response to NACT and clinical stage (φ = 0.4365), the clinical stage and tumor size (φ = 0.5789), and the clinical stage and lymph node status (φ = 0.6395).

There is a moderate correlation between the NACT response and lymph node status in the luminal B group of tumors (*p* < 0.05; φ = 0.573). In addition, a high correlation was found in the luminal B group between the clinical stage of the tumor and its size (*p* < 0.05; φ = 0.7976).

The association of the clinical stage with the tumor size and the lymph node status may be explained by the derivation of the clinical stage from the TNM characterization of the tumor.

The results of the correlation analysis suggest that the addition of the variable “clinical stage, S” for TN breast cancer, and “lymph node status, N” for luminal B tumors might add to the quality of the developed molecular epigenetic classifiers. We have added S and N predictors to DNA methylation markers for TN and luminal B tumors, respectively, and reevaluated the diagnostic characteristics of the resulting panels (Appendix A and Figure 5).

Using classifiers, consisting not only of epigenetic markers, but also of clinical ones, makes it possible to develop a classifier with significantly better characteristics than using only DNA methylation markers or clinical features (Figure 6). For lum.B tumors, a DNA methylation panel including “*LTBR*, *CLEC14A*, N” shows cvAUC = 0.83; 95% CI = 0.82–0.85, with a sensitivity of 0.89 and a specificity of 0.71. For TN breast cancer, the inclusion of the clinical stage in the epigenetic classifier made it possible to achieve an area under the curve cvAUC = 0.87; 95% CI = 0.86–0.88 with a sensitivity of 0.71 and a specificity of 0.80 for the “*TMEM132D*, *TMEM132C*, *MYO15B*, S” panel.

We assessed epigenetic and combined marker panels using a likelihood ratio test. As a result, the combined model for TN breast cancer (*TMEM132D*, *TMEM132C*, *MYO15B*, S) demonstrated statistical significance (*p* = 0.0004) vs. the epigenetic model (*TMEM132D*, *TMEM132C*, *MYO15B*). The luminal B breast cancer subtype combined (*LTBR*, *CLEC14A*, N) vs. the epigenetic model (*LTBR*, *CLEC14A*) also showed statistical significance (*p* = 0.006).

To investigate the potential predictive value of combined epigenetic and clinical panels, we calculated the NACT response score for epigenetic panels developed within this study and combined it with independent clinical features, such as the tumor size, lymph node involvement, clinical stage, and patient’s age (Figure 7). The association between the NRS and NACT response retained statistical significance when using clinical features, such as the clinical stage for TN breast cancer subtype (OR = −0.47 (95% CI = −0.76; −0.20) *p* = 0.002) and lymph node involvement for luminal B breast cancer (OR = −0.55 (95% CI = −0.86; −0.24), *p* = 0.001).

## 4. Discussion

Breast cancer is a group of diseases that is extremely heterogeneous in terms of clinical and morphological characteristics, progression, survival, response to therapy, and molecular profiling [18]. For BC, molecular genetic classifiers are being actively developed with the prospect of using them as sets of prognostic and predictive markers [19]. Although BC is among the most exhaustively studied cancers, there is still a lack of molecular markers to predict its response to neoadjuvant chemotherapy (NACT) [20]. Tumor development is determined by alterations not only of its genome, but of its epigenome as well. In order to identify epigenomic markers of BC NACT effectiveness, we have applied genome-wide DNA methylation screening of tumors in cohorts of NACT responders and nonresponders. As a result, we demonstrate the predictive potential of DNA methylation markers in assessing the BC response to NACT. To characterize DNA methylation markers in an independent cohort, the MSRE-qPCR method was used, which has competitive analytical sensitivity and allows a reduction in analysis time and expenses, as well as the volume of input tissue material, which is critical in biopsy analysis.

We have performed genome-wide bisulfite sequencing on 73 BC biopsy samples (discovery cohort) obtained prior to NACT to select genome regions to further develop DNA methylation panels of a limited number of markers. The discovery cohort consisted of 29 samples of TNBC and 44 samples of luminal B tumors. Other research groups have developed sets of genome-wide DNA methylation data, some on larger collections of BC samples [21,22,23]; however, the most common way to obtain data nowadays is probe hybridization that allows for determining the status of the methylation of only a predesigned selection of CpG dinucleotides. The fundamental difference between the methods based on microarray hybridization and bisulfite sequencing is that the latter allows an assessment of each and every CpG pair in a locus, thus presenting the full picture of methylation at a genome regulatory region of interest.

To perform genome-wide bisulfite sequencing, we have used our version of reduced representation bisulfite sequencing, XmaI-RRBS [12]. In general, compared to the whole-genome bisulfite sequencing (WGBS), RRBS approaches increase the relative informational value of DNA methylation analysis: each RRBS sequence read includes at least one informative CpG position. Theoretically, RRBS is better applicable to large-scale studies of DNA methylation and in the search for markers of epigenetic processes in health and disease than WGBS, since RRBS focuses on functionally significant methylation in CpG islands, which constitute a small portion of the genome, and ignores the major portion that is less significant for clinical research [9].

To our knowledge, by now only one panel of DNA methylation markers to predict the BC response to NACT have been published [24]. Begona Pineda et al. have proposed a DNA methylation assay based on the levels of methylation of the *FERD3L* and *TRIP10* genes, with AUC = 0.905 (78.6% accuracy), for predicting pathological complete response (pCRs) in TNBC patients treated with anthracyclines and taxanes [24]. In their study, exploratory analysis for a genome-wide search for DNA methylation markers was performed using Infinium Human Methylation 450 K microarrays on a cohort of 24 patients, twice smaller than ours. However, it is valuable that their study was the first to demonstrate the potential of DNA methylation markers in assessing the BC response to NACT.

We previously published a study [9] in which we examined 25 luminal B breast cancer biopsies obtained prior to neoadjuvant chemotherapy. In the current study, these 25 samples were included in a bigger discovery cohort and the fundamental approach to data analysis and the formation of panels from a limited number of DNA methylation markers was changed. For example, we selected markers among loci for which methylation information was obtained for at least four consecutive CpG pairs. The change in approach to the analysis of XmaI-RRBS results does not allow us to combine the results of the two studies; however, the expanded discovery sample provides a broader view of differential methylation in the luminal B breast cancer.

Some of the predictive factors used to make decisions regarding BC systemic treatment are the tumor size, status of lymph node involvement, and clinical stage [25]. Moreover, in addition to the widely used surrogate molecular markers ER, PR, HER2, and Ki-67, attempts are being made to search for both new potential markers and a more detailed study of markers that are not common in clinical practice, such as proliferating cell nuclear antigen PCNA [26], caveolin [27], and chemokine receptor CXCR4 [28]. Since there is still no gold standard, it is important to continue attempts to search for new predictive markers and, if possible, to pay attention to combined panels of markers of various natures (genetic, epigenetic, transcriptomic, proteomic, clinical, morphological).

Some of the epigenetic markers identified in our study, *TMEM132D*, *MYO15B*, *TTC34*, *LTBR,* and *CLEC14A* genes, whose methylation levels showed the best classification of BC sensitivity to NACT, were previously partially studied in terms of involvement in the etiopathogenesis of malignant neoplasms and potential use as diagnostic markers in oncology.

Mutations in the *TMEM132D* gene were found in pancreatic cancer [29] and small cell lung cancer [30]. The gene itself codes for transport receptors in the brain. Specific studies of its methylation in breast cancer, or associations with chemotherapy, have not been performed, but it was published that its overexpression correlates with cytotoxic T-lymphocytes infiltration and better survival in patients with early-stage ovarian cancer [31]. *MYO15B* genetic variants were found to be associated with an increased risk of depressive disorder in females [32], and the function of the protein is unknown, since it lacks a motor domain according to the GeneCards database. *TTC34* is a paralog of the *TMTC4* gene responsible for protein–protein interactions. The function of TTC34 is unknown. *TTC34* was upregulated in luminal and triple-negative BC subtypes upon LINC01087 overexpression [33]. *LTBR* (lymphotoxin beta receptor) is probably the most interesting of the list. It is a member of the tumor necrosis factor family and is closely associated with carcinogenesis [34], but gene methylation has not previously been studied as a predictive marker in oncology. *CLEC14A* is involved in regulating the growth of cancer cells, maintaining body hemostasis, and facilitating cell communication [35]. It is found in the tumor endothelium and is a tumor endothelial marker [36]. Gene methylation is also associated with carcinogenesis [37] but has not been studied as a predictive marker in breast cancer.

Finally, some limitations and perspectives of this study are to be discussed. Many of the markers selected in the discovery cohort are not significantly associated with the NACT response in the independent cohort. This can be explained both by the use of two fundamentally different platforms for the analysis of cohorts and by the use of a nominal *p*-value when selecting markers from the results of the genome-wide RRBS screening, which could lead to false positive hits included in the list of candidate markers identified in the discovery cohort. Taking into account such limitations, we formed a deliberately large number of panels for validating RRBS findings and formed both subtype-specific test systems and universal ones (for TN and lum.B subtypes of breast cancer).

In order to support our results, larger independent cohorts should be tested. This will increase precision and allow us to better understand the meaning of DNA methylation in terms of the responses of different subtypes of BC to NACT. If our findings are well confirmed on independent validation cohorts and lead to the clinical implementation of DNA methylation markers for treatment prediction, an improvement in BC NACT may be anticipated. One of the intriguing research perspectives is the assessment of the dynamics of methylation levels of the markers as a result of NACT. Studying methylation changes associated with chemotherapy may shed more light on the biology of the tumor response to treatment and on the nature of tumor chemoresistance.

## 5. Conclusions

As a result of this study, we confirmed our hypothesis about the high predictive potential of DNA methylation markers in assessing the response to neoadjuvant chemotherapy in breast cancer. The use of combined predictors, including not only epigenetic markers, but also data obtained in the course of clinical and morphological examinations, allows us to approach the issue of personalized chemotherapy prescription for patients with breast cancer. Since there is still no gold standard, it is necessary to continue attempts to search for new predictive markers and, possibly, pay attention to complex panels of markers of various natures (genetic, epigenetic, proteomic, clinical, morphological).

## Figures and Tables

**Figure 1 cancers-15-01630-f001:**
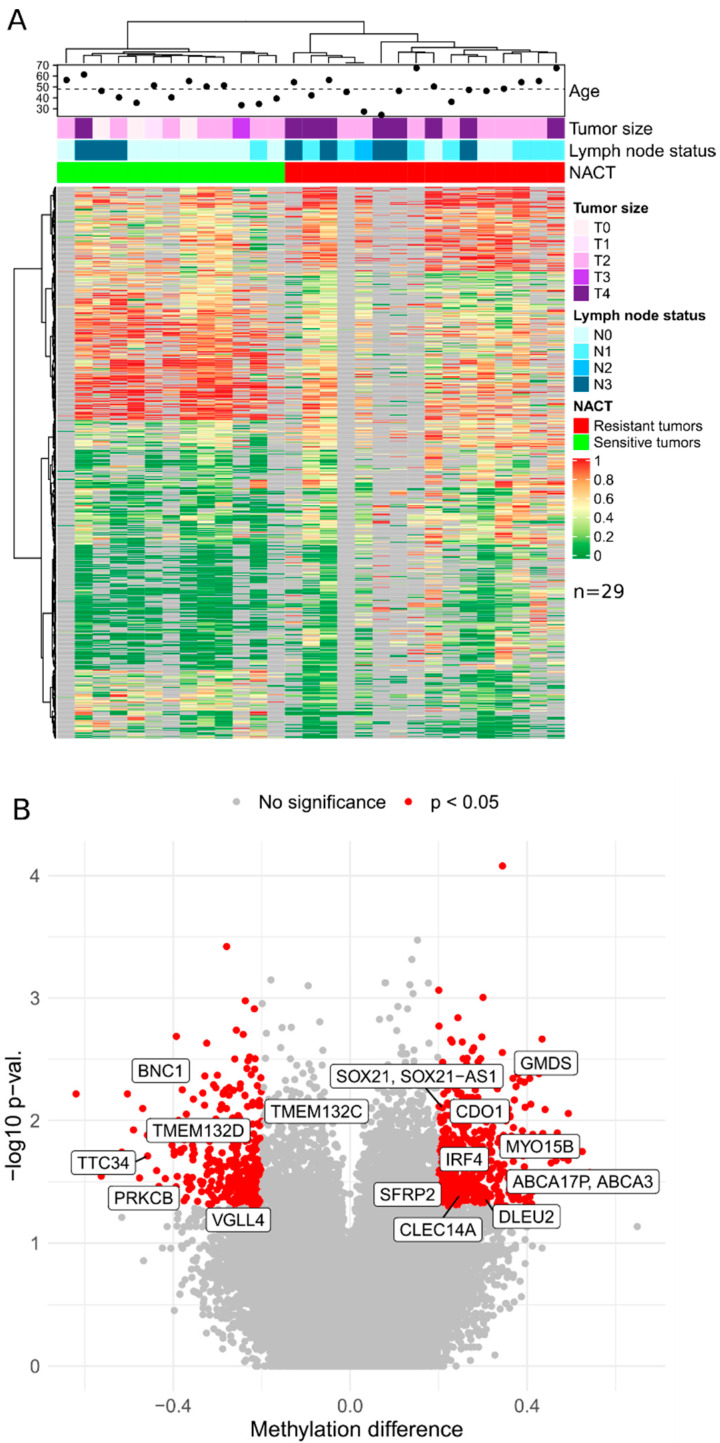
(**A**) Genome-wide bisulfite DNA sequencing (XmaI-RRBS) results from TNBC samples obtained before neoadjuvant chemotherapy. The heat map shows the level of methylation: red—100%, yellow—50%, green—0%, gray—missing data in NACT-sensitive and -resistant TNBC samples. Each row represents the methylation of a CpG dinucleotide in the samples by column. (**B**) Volcano plot indicating differentially methylated CpG-dinucleotides in TNBC tumors. X axis represents methylation difference between resistant and sensitive tumor samples; Y axis stands for −log10 scaled nominal *p*-value. Most informative CpG-pairs revealed by MSRE-qPCR in an independent cohort are marked with gene names they map to.

**Figure 2 cancers-15-01630-f002:**
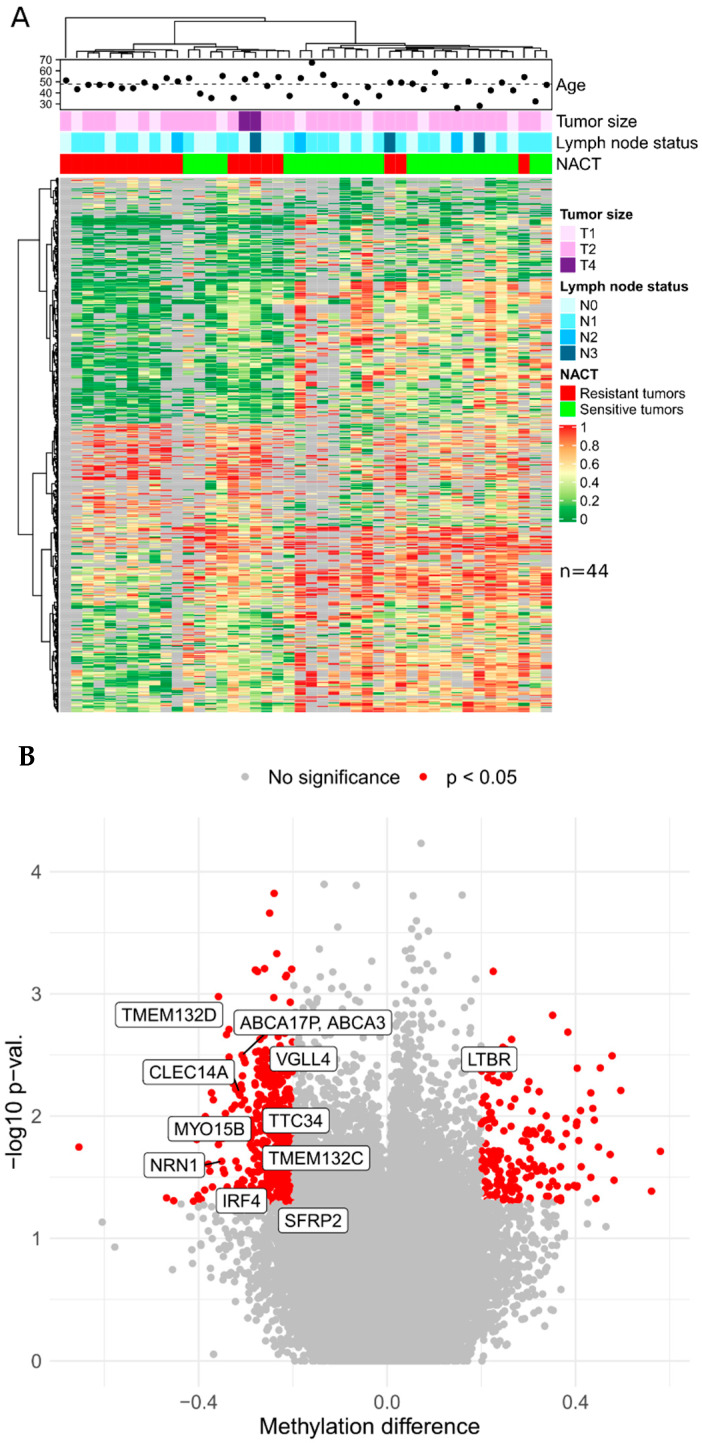
(**A**) Genome-wide bisulfite DNA sequencing (XmaI-RRBS) results from luminal B breast cancer samples obtained before neoadjuvant chemotherapy. The heat map shows the level of methylation: red—100%, yellow—50%, green—0%, gray—missing data in NACT-sensitive and -resistant luminal B breast cancer samples. Each row represents the methylation of a CpG dinucleotide in the samples by column. (**B**) Volcano plot indicating differentially methylated CpG-dinucleotides in luminal B tumors. *X*-axis represents methylation difference between resistant and sensitive tumor samples; *Y*-axis stands for −log10 scaled nominal *p*-value. Most informative CpG-pairs revealed by MSRE-qPCR in an independent cohort are marked with gene names they map to.

**Figure 3 cancers-15-01630-f003:**
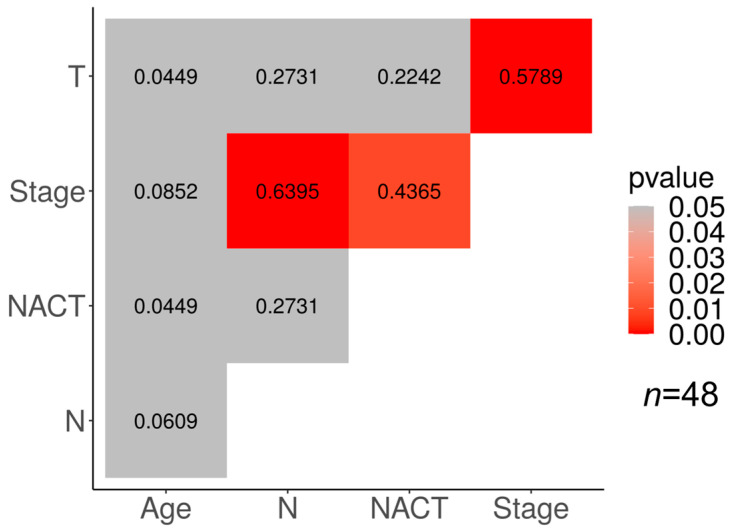
Relationship between clinical characteristics and response to neoadjuvant chemotherapy in TNBC. Significant correlations are marked in red (*p* < 0.05); numbers show the correlation coefficient.

**Figure 4 cancers-15-01630-f004:**
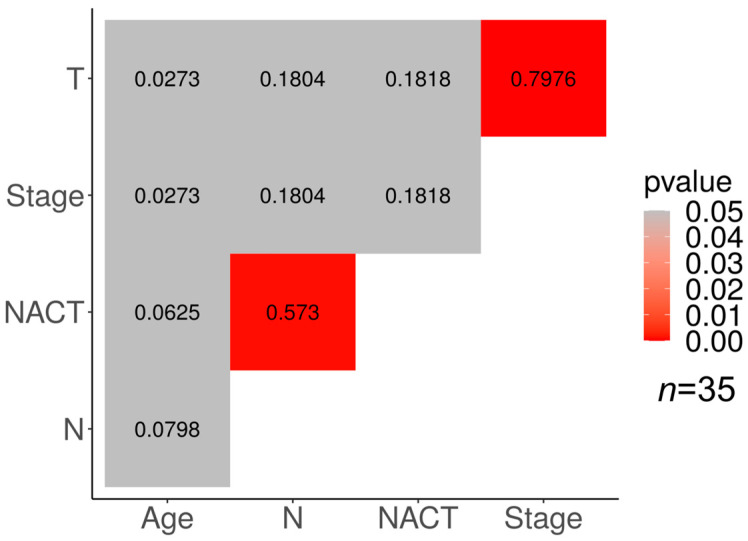
Relationship between clinical characteristics and response to neoadjuvant chemotherapy in luminal B breast cancer. Significant correlations are marked in red (*p* < 0.05); numbers show the correlation coefficient.

**Figure 5 cancers-15-01630-f005:**
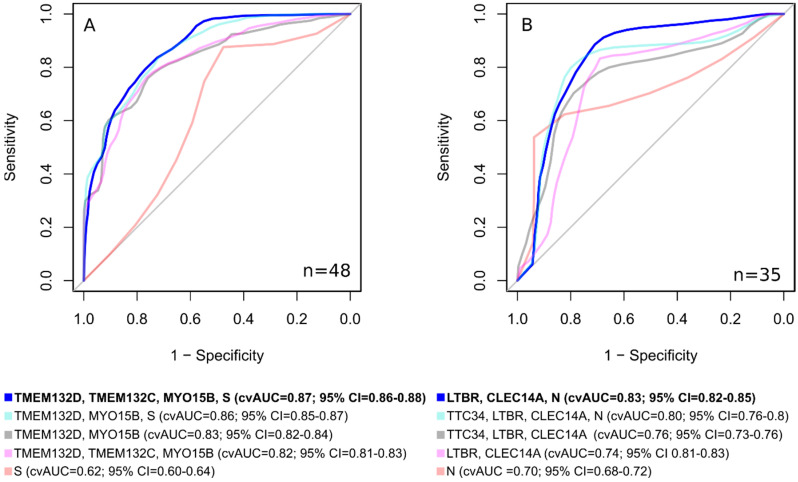
ROC curves, cross-validated area under the curve (cvAUC) for breast cancer NACT sensitivity classifiers for TN (**A**) and luminal B subtypes (**B**), consisting of separate clinical, separate molecular epigenetic, and combined predictors.

**Figure 6 cancers-15-01630-f006:**
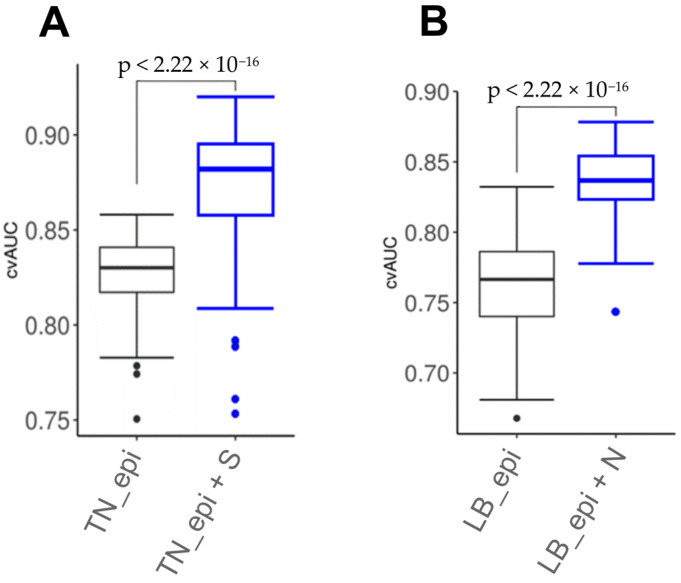
Boxplots for breast cancer NACT sensitivity classifiers, consisting of epigenetic (_epi) and combined epigenetic and clinical predictors (_epi + S, _epi + N) for (**A**) TN and (**B**) luminal B (LB) breast tumors.

**Figure 7 cancers-15-01630-f007:**
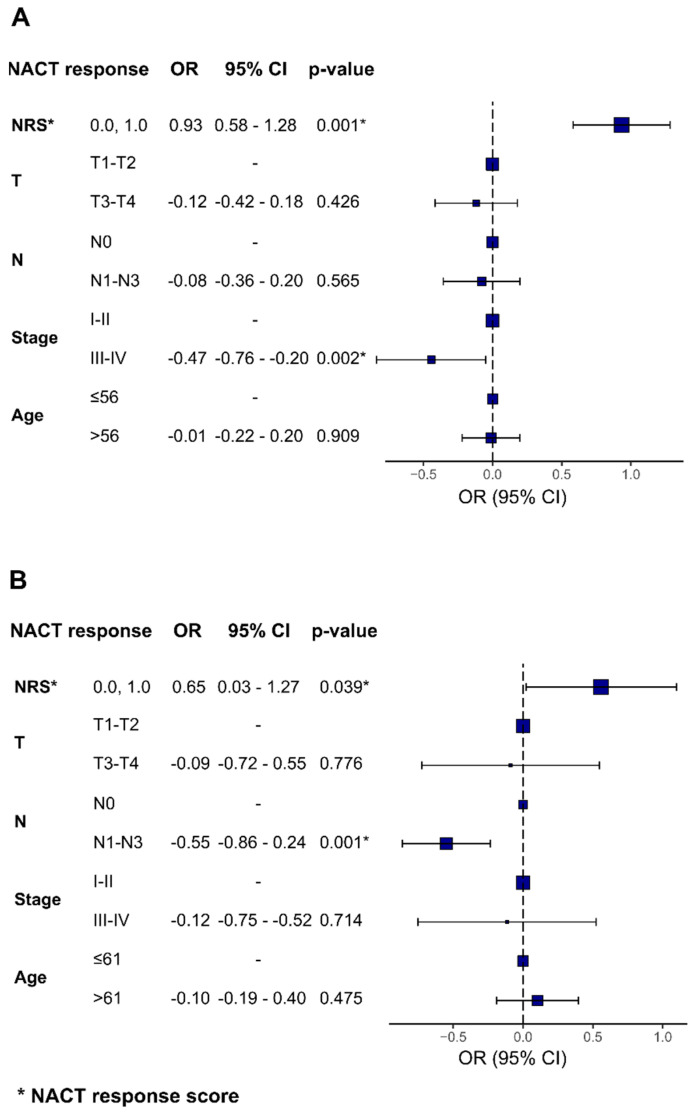
Forest plots indicating the odds ratios for TN tumors’ (**A**) and luminal B tumors’ (**B**) NACT response and the corresponding confidence intervals. The higher the odds ratio for NRS, the more likely the patient will demonstrate a positive response to therapy.

**Table 1 cancers-15-01630-t001:** Clinical and pathological characteristics of discovery cohort patients (*n* = 73) and independent cohort patients (*n* = 83). *p*-value shows statistical differences in clinical features between cohorts.

		Discovery Cohort		Independent Cohort		*p*-Value ^1^ TN BC	*p*-Value ^2^ lum.B BC
		TNBC, N (%); *n* = 29	Luminal B, N (%); *n* = 44	TNBC, N (%); *n* = 48	Luminal B, N (%); *n* = 35		
Age median		48 (min 25; max 68)	48 (min 27; max 68)	56 (min 34; max 77)	61 (min 37; max 75)	0.02	<0.001
Tumor size	T0	3 (10.3)	-	-	-	0.76	0.06
	T1	1 (3.4)	10 (22.7)	3 (6.25)	-		
	T2	15 (51.7)	32 (72.2)	30 (62.5)	28 (80)		
	T3	1 (3.4)	-	6 (12.5)	6 (17.14)		
	T4	9 (31)	2 (4.5)	9 (18.75)	-		
	No data	-	-	-	1 (2.86)		
Lymph node status	N0	12 (41.4)	16 (36.4)	29 (60.42)	12 (34.29)	0.10	0.99
	N1	8 (27.6)	22 (50.0)	10 (20.83)	22 (62.86)		
	N2	1 (3.4)	3 (6.8)	2 (4.17)	-		
	N3	8 (27.6)	3 (6.8)	7 (27.6)	-		
	No data	-	-	-	1 (2.86)		
NACT response	Sensitive tumors	13 (44.8)	25 (56.8)	33 (68.75)	19 (55)	0.03	0.82
	Resistant tumors	16 (55.2)	19 (43.2)	15 (31.25)	16 (45)		

^1^—*p*-value for comparing clinical features between cohorts in TN breast cancer. ^2^—*p*-value for comparing clinical features between cohorts in luminal B breast cancer.

**Table 2 cancers-15-01630-t002:** Methylation marker genes selected for MSRE-qPCR assays and the function of proteins encoded by these genes.

Pool (Test-System)	Target Locus	Related Function of the Encoded Proteins
G1	TERT	Cellular regulation
	TTC34_1	Transmembrane transporters and their regulators
	TMEM132D_3	Transmembrane transporters and their regulators
G2	TMEM132D_2	Transmembrane transporters and their regulators
	VGLL4_2	Transcription regulation
G3	ABCA3_1	Transmembrane transporters and their regulators
	DPYS_1	Metabolic enzymes and their regulators
	IRF4	Immune response
G4	TMEM132C_1	Transmembrane transporters and their regulators
	SFRP2_1	Receptors and signal transduction
	SOX21	Transcription regulation
G5	MYO15B	Noncoding transcripts
	TMEM132D	Transmembrane transporters and their regulators
TN1	ABCA3_1	Transmembrane transporters and their regulators
	CDO1_1	Metabolic enzymes and their regulators
	CLEC14A_1	Cell adhesion
TN2	DLEU2_1	Noncoding transcripts
TN3	BNC1_1	Transcription regulation
	SFRP2_1	Receptors and signal transduction
	TTC34_1	Transmembrane transporters and their regulators
TN4	PRKCB_2	Receptors and signal transduction
	GMDS_1	Metabolic enzymes and their regulators
	PRKCB_2	Receptors and signal transduction
TN6	MYO15B_1	Noncoding transcripts
	TMEM132D_1	Transmembrane transporters and their regulators
LB1	LTBR	Receptors and signal transduction
	NRN1	Cellular regulation

## Data Availability

The sequencing data obtained are placed in the GEO database, under the numbers GSE123712 and GSE123828, and can be used by other research teams to conduct comparative studies and work on the study of breast cancer epigenomics.

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
