# Peer review of "DNA Methylation and Prospects for Predicting the Therapeutic Effect of Neoadjuvant Chemotherapy for Triple-Negative and Luminal B Breast Cancer"

_cancers, 2023, doi:10.3390/cancers15051630_

Round 1
Reviewer 1 Report (Previous Reviewer 2)
This reviewer's prior recommendations have been addressed.
Please add NRS in full to the legend of the new Figure 7.
No new concerns.
Author Response
Response to Reviewer 1 Comments
Point 1: Please add NRS in full to the legend of the new Figure 7.
No new concerns.
Response 1: Thank you for this comment. We added NRS (under the asterisk) in full to the Figure 7.
We thank Reviewer 1 for the extensive and careful work that was done for improving the quality of our manuscript.

Reviewer 2 Report (New Reviewer)
In this work Sigin et al investigated the role of methylation in determining response to NAC in TNBC and Luminal B breast tumors. The scientific question is very interesting, and they also applied a technological approach that could be transferred to the clinical easier than classical whole genome bisulfite sequencing. Despite the results show potential informative role of methylation to predict NAC response, my biggest concern is about the statistical robustness of the study since small sample size are used and some methodological aspects can be improved.
Below there are a series of comments to the presented work.
I suggest merging Table 1 and Table 2 and add columns of p-values for the comparison of discovery and independent cohort for the two subtypes. This would make easier to compare the clinical characteristics of the two cohorts.
Section 2.8 of materials and methods: what the authors mean with “To determine the structure of correlations between the methylation level of epigenetic markers and clinical/morphological features, the coefficient ρ was used” and “To determine the structure of the relationship between the response to therapy and clinical/morphological features, the coefficient φ was used”? What is intended for the “structure” of the association? And what are coefficients ρ and φ. Please describe better the methodology and the purpose of these analyses, with appropriate references if needed.
Authors used correlation to test the independence between DNA methylation markers and the clinical parameters of patients. Despite correlation provides an idea of the association between two continuous variables it would be better to use a formal statistical test such as using a multivariable model (such as logistic regression) including for each gene methylation data and clinical parameters. This would give a formal assessment of the informativeness and independence of the variables.
Authors selected candidate markers according to a p < 0.05. It is not clear if this p-value is the nominal p-value or the adjusted p-value. Please clarify. For completeness authors could add a volcano plot in Figure 1 and Figure 2 highlighting the number of differentially methylated genes after p-value adjustment. If no genes pass the significance threshold after p-value adjustment authors should comment on this.
Figure 1 and 2. What indicates grey color? NAs? Please explain in the figure legend. Would be useful to add clinical annotation at the top of the heatmap to visually inspect their distribution according to the methylation pattern.
Supplementary material: The list of differentially methylated genes should be presented as a table with quantitative information about the direction of the differential methylation (fold change), nominal p-values, and adjusted p-values. In addition, the authors state that TERT and DPYS were not significantly associated to response, but they were selected for their biological relevance. In reason of this these two genes should not be inserted in these lists.
Are candidate markers enriched for specific biological functions? Add an over-representation analysis to test this.
Lines 329 and 333. You are not testing the diagnostic value but the predictive value of the marker. Please revise here and throughout the text
Section 3.3 of the results: it is not clear how the authors combined markers in multivariable panels and how they performed the cvAUC calculation using them? Since logistic regression is cited in the materials and methods, I assume they used this method, but this section should be expanded and better explained.
Section 3.4. It is difficult to evaluate the additive informative value of S and N to the epigenetic models since molecular only and combined models are different (for example, in TN the combined model includes also TMEM132C that is missing from the molecular only model. Authors should show only the combined and the molecular model with the same molecular features. In addition, the informativeness of the clinical variables should be formally assessed using a statistical test (e.g likelihood ratio test between the molecular and the combined model).
Results from lines 404 to 411 and figure 7. In my opinion this analysis does not add information to the manuscript since it was already described that only stage for TN and lymph node status for LumB are associated with response. It can be removed.
Discussion line 434 “Other research groups have developed sets of genome-wide DNA methylation data”: add references
Many of the markers selected in the discovery cohort are not significantly associated to response in the independent cohort. This is probably due to the many false positive hits included in the list of candidate markers identified in the discovery cohort, since selection was based on nominal p-value (i think, see my previous comment). Authors should discuss more about this and the statistical limitations of their study in the discussion session.
The authors already published a paper in which they identified predictive methylation markers of NAC response in Luminal B breast cancer (ref 9). Why the authors did not use this data to validate their model for luminal B tumors. Are the samples already published included in their new cohort under analyses. It is also interesting to note that of their 15 candidate markers identified in ref 9 only four (IRF4 SNAP25 PRR5 PTGIS) overlap with the list of 200 markers selected in this work (I did not consider TERT and DPYS). Moreover, only IRF4 was included in the selected marker panel used in the independent cohort in this work. This is indicative of a lack of robustness in the results due possibly related to a lack of strong statistical power, since the selected cohorts of patients are relatively small. The authors should discuss their results in relation to their previous findings.
In another work Kong et al (https://doi.org/10.3892/ol.2020.11737) showed that PCDH17 is associated to NAC response in TNBC. Is this marker present in the data of the authors? If yes, is it associated to response in their cohort?
Author Response
Response to Reviewer 2 Comments
Point 1: I suggest merging Table 1 and Table 2 and add columns of p-values for the comparison of discovery and independent cohort for the two subtypes. This would make easier to compare the clinical characteristics of the two cohorts.
Response 1: Thank you very much for this comment. We have merged Table 1 and Table 2 and added columns of p-values.
Point 2: 2.8 of materials and methods: what the authors mean with “To determine the structure of correlations between the methylation level of epigenetic markers and clinical/morphological features, the coefficient ρ was used” and “To determine the structure of the relationship between the response to therapy and clinical/morphological features, the coefficient φ was used”? What is intended for the “structure” of the association? And what are coefficients ρ and φ. Please describe better the methodology and the purpose of these analyses, with appropriate references if needed.
Response 2: Thank you. We removed “structure” and added a detailed description of the coefficients used to better describe the statistical processing.
Point 3: Authors used correlation to test the independence between DNA methylation markers and the clinical parameters of patients. Despite correlation provides an idea of the association between two continuous variables it would be better to use a formal statistical test such as using a multivariable model (such as logistic regression) including for each gene methylation data and clinical parameters. This would give a formal assessment of the informativeness and independence of the variables.
Response 3: We have used a point-biserial correlation which allows to resolve the issue regarding the independence of markers with clinical parameters (between continuous and categorical variables).
Point 4: Authors selected candidate markers according to a p < 0.05. It is not clear if this p-value is the nominal p-value or the adjusted p-value. Please clarify. For completeness authors could add a volcano plot in Figure 1 and Figure 2 highlighting the number of differentially methylated genes after p-value adjustment. If no genes pass the significance threshold after p-value adjustment authors should comment on this.
Response 4: Thank you for this comment. We used nominal p-value and clarified it in manuscript. We added volcano plots in Figure 1 and Figure 2. We have discussed using nominal p-value in limitations part of the Discussion.
Point 5: Figure 1 and 2. What indicates grey color? NAs? Please explain in the figure legend. Would be useful to add clinical annotation at the top of the heatmap to visually inspect their distribution according to the methylation pattern.
Response 5: Thank you very much. We explained missing data (grey color) in figures description and added clinical annotations at the top of the heatmaps.
Point 6: Supplementary material: The list of differentially methylated genes should be presented as a table with quantitative information about the direction of the differential methylation (fold change), nominal p-values, and adjusted p-values. In addition, the authors state that TERT and DPYS were not significantly associated to response, but they were selected for their biological relevance. In reason of this these two genes should not be inserted in these lists.
Response 6: Thank you. We added tables of differentially methylated genes with quantitate information in supplementary table S10 (without TERT and DPYS genes).
Point 7: Are candidate markers enriched for specific biological functions? Add an over-representation analysis to test this.
Response 7: Thank you for this comment. We added over-representation analysis with KEGG and GO databases (Supplementary Figures S3, S4, S5, Supplementary Table S11).
Point 8: Lines 329 and 333. You are not testing the diagnostic value but the predictive value of the marker. Please revise here and throughout the text.
Response 8: Thank you for this comment. We have corrected all errors in the document and supplementary materials.
Point 9: Section 3.3 of the results: it is not clear how the authors combined markers in multivariable panels and how they performed the cvAUC calculation using them? Since logistic regression is cited in the materials and methods, I assume they used this method, but this section should be expanded and better explained.
Response 9: Thank you. We have added a more detailed description in the Materials and Methods section.
Point 10: Section 3.4. It is difficult to evaluate the additive informative value of S and N to the epigenetic models since molecular only and combined models are different (for example, in TN the combined model includes also TMEM132C that is missing from the molecular only model. Authors should show only the combined and the molecular model with the same molecular features. In addition, the informativeness of the clinical variables should be formally assessed using a statistical test (e.g likelihood ratio test between the molecular and the combined model).
Response 10: Thank you for comment. Having added independent clinical variables, we rerun the analysis to reveal combined panels (this is now reflected in the text of the manuscript). Additional ROC curves were applied to the Figure 5, including all additional panels.
Point 11: Results from lines 404 to 411 and Figure 7. In my opinion this analysis does not add information to the manuscript since it was already described that only stage for TN and lymph node status for LumB are associated with response. It can be removed.
Response 11: Thank you. Adding such an analysis and a corresponding figure was suggested by another reviewer, so we choose to retain this part.
Point 12: Discussion line 434 “Other research groups have developed sets of genome-wide DNA methylation data”: add references.
Response 12: Thank you. We added references.
Point 13: Many of the markers selected in the discovery cohort are not significantly associated to response in the independent cohort. This is probably due to the many false positive hits included in the list of candidate markers identified in the discovery cohort, since selection was based on nominal p-value (i think, see my previous comment). Authors should discuss more about this and the statistical limitations of their study in the discussion session.
Response 13: Thank you very much. We have added a paragraph to the discussion section.
Point 14: The authors already published a paper in which they identified predictive methylation markers of NAC response in Luminal B breast cancer (ref 9). Why the authors did not use this data to validate their model for luminal B tumors. Are the samples already published included in their new cohort under analyses. It is also interesting to note that of their 15 candidate markers identified in ref 9 only four (IRF4 SNAP25 PRR5 PTGIS) overlap with the list of 200 markers selected in this work (I did not consider TERT and DPYS). Moreover, only IRF4 was included in the selected marker panel used in the independent cohort in this work. This is indicative of a lack of robustness in the results due possibly related to a lack of strong statistical power, since the selected cohorts of patients are relatively small. The authors should discuss their results in relation to their previous findings.
Response 14: Thank you for this comment and careful reading of our previous publications. We have added a paragraph to the discussion section.
Point 15: In another work Kong et al (https://doi.org/10.3892/ol.2020.11737) showed that PCDH17 is associated to NAC response in TNBC. Is this marker present in the data of the authors? If yes, is it associated to response in their cohort?
Response 15: Thank you. Unfortunately, PCDH17 was not present among our RRBS targets, so we cannot evaluate its association with NACT response.
We thank Reviewer 2 for the extensive and careful work that was done for improving the quality of our manuscript.

Reviewer 3 Report (New Reviewer)
Dr. Sigin et al. screened DNA methylation markers for predicting the response of triple-negative and luminal-B breast cancers to neoadjuvant chemotherapy. This study was likely well designed. However, additional analyses are needed.
1. They used a digestion control in the quantitative methylation analysis based on methylation-sensitive restriction. However, when the results of an analysis should be excluded from final analysis was not mentioned in the method section.
2. They established two prediction panels of methylation markers for 48 patients with the triple-negative breast cancer and 35 patients with luminal-B breast cancer in the independent cohort. It is needed to validate the performance of these panels using these samples from 73 patients in the discovery cohort. At least they need to validate the performance using the RRBS results for the patients in the discovery cohort.
3. As patients administrated with neoadjuvant chemotherapy, surgical samples were available for the majority of patients. It will be useful for readers if they would compare the stability of methylation changes of target CpG sites in samples before and post neoadjuvant chemotherapy.
4. Breast cancer are well studied cancer. There are many publicly available genome-wide DNA methylation resources. To increase the reproducibility of their conclusion, the reviewer suggests they perform some bioinformatic analysis using these DNA methylation datasets.
Author Response
Response to Reviewer 3 Comments
Point 1: They used a digestion control in the quantitative methylation analysis based on methylation-sensitive restriction. However, when the results of an analysis should be excluded from final analysis was not mentioned in the method section.
Response 1: Thank you very much for this comment. We have added a more complete description in the material and methods section. DC scores were not used for target methylation level calculation. They were only used for DNA hydrolysis quality control. Samples that demonstrated ΔCt < 6 for DC in digested versus nondigested aliquots were excluded from final analysis.
Point 2: They established two prediction panels of methylation markers for 48 patients with the triple-negative breast cancer and 35 patients with luminal-B breast cancer in the independent cohort. It is needed to validate the performance of these panels using these samples from 73 patients in the discovery cohort. At least they need to validate the performance using the RRBS results for the patients in the discovery cohort.
Response 2: Thank you for comment. We added cvAUCs for breast cancer NACT sensitivity classifiers validated using discovery cohort RRBS data (Supplementary Figure S6).
Point 3: As patients administrated with neoadjuvant chemotherapy, surgical samples were available for the majority of patients. It will be useful for readers if they would compare the stability of methylation changes of target CpG sites in samples before and post neoadjuvant chemotherapy.
Response 3: This is a very intriguing topic to discover indeed, and we are working on it now. Yet, it is a separate study not entirely related to predicting the effect of neoadjuvant chemotherapy which is the point of the current manuscript. We plan to present an article on methylation changes of CpG sites before and post neoadjuvant chemotherapy later this year.
Point 4: Breast cancer are well studied cancer. There are many publicly available genome-wide DNA methylation resources. To increase the reproducibility of their conclusion, the reviewer suggests they perform some bioinformatic analysis using these DNA methylation datasets.
Response 4: Thank you for this comment. Unfortunately, public breast cancer datasets rarely contain information about neoadjuvant therapy and response to it. In addition, almost all studies were carried out on DNA methylation microarrays (we discussed this in the text of the manuscript). The only public METABRIC dataset (https://doi.org/10.1038/s41467-021-25661-w) that contains information about NACT is technically problematic to access at present.
We thank Reviewer 3 for the extensive and careful work that was done for improving the quality of our manuscript.

Round 2
Reviewer 2 Report (New Reviewer)
The authors answered adequately to the requests of this reviewer. There is an overall improvement of the quality of their work.
I have only a minor suggestion: throughout the text luminal B is sometimes written as "luminal B", sometime as "lum.B". Please make the nomenclature uniform for consistency.
Reviewer 3 Report (New Reviewer)
No more comment.
This manuscript is a resubmission of an earlier submission. The following is a list of the peer review reports and author responses from that submission.
Round 1
Reviewer 1 Report
With breast cancer cases of 29 triple negative breast cancer (TNBC) and 44 luminal B, differentially methylated genes related to neoadjuvant chemotherapy response were screened by XmaI-RRBS. MSRE-qPCR method was employed for validation in the same cases of breast cancer. Authors argued that methylation in TMEM123D and MYO15B for TNBC and TTC34, LTBR and CLEC14A for luminal B showed high cross-validated area under the ROC curve (0.83 and 0.76, respectively) probably with 48 TNBC and 35 luminal B cases (not clearly presented in the manuscript).
However, the following results have not been presented in the manuscript
1) In the screening phase, the statistical significance or P value of the markers or marker combinations on the chemotherapy response has not been presented in the manuscript.
2) In the validation phase, the number of cases for MSRE-qPCR has not been described in the results section (the case numbers should be presented in the results section and in each figure), and the significance of markers on the chemotherapy response has not been presented.
3) After separating the study group depending on markers or marker combinations, Kaplan-Meier plot and multivariate analysis using T and N stages as co-variants should be presented. The figure 8 in the manuscript may not be a valid presentation.
Minor comment
1) As indicated in the discussion section, epigenetic markers such as FERD3L or TRIP10 in TNBC cases have been identified previously. Those markers can be evaluated with the authors’ analyses, and the description of their significance may help other researchers.
2) Figures 3 and 4 should be improved. It is hard to understand what the authors want to argue in those figures.
Author Response
Point 1: In the screening phase, the statistical significance or P value of the markers or marker combinations on the chemotherapy response has not been presented in the manuscript.
Response 1: Thank you for this comment. We added statistical significance in “DNA methylation markers selection criteria” section.
Point 2: In the validation phase, the number of cases for MSRE-qPCR has not been described in the results section (the case numbers should be presented in the results section and in each figure), and the significance of markers on the chemotherapy response has not been presented.
Response 2: Thank you. We have added corrections to each figure and to the “Results” section. Significance of markers was added in Supplementary tables S4 and S5.
Point 3: After separating the study group depending on markers or marker combinations, Kaplan-Meier plot and multivariate analysis using T and N stages as co-variants should be presented. The figure 8 in the manuscript may not be a valid presentation.
Response 3: Thank you for your comment. Unfortunately, we do not have any information on patient survival, so we cannot perform these analyzes. However, given the independence of clinical features, we consider it possible to add them to the classifiers using ROС-analysis with cross-validation as independent variables.
Point 4: As indicated in the discussion section, epigenetic markers such as FERD3L or TRIP10 in TNBC cases have been identified previously. Those markers can be evaluated with the authors’ analyses, and the description of their significance may help other researchers.
Response 4: Thank you very much for the comment. CpG pairs from the study under discussion were not included in our RRBS analysis, as it targets CpG islands (using a specific restriction enzyme XmaI). Probes from the Peneda’s et al. study hit areas outside the CpG-islands.
Point 5: Figures 3 and 4 should be improved. It is hard to understand what the authors want to argue in those figures.
Response 5: Thank you for this comment. In order not to overload the reader with large figures that do not demonstrate any statistical significance, we transferred them to supplementary materials (Figures S1 and S2).
We thank Reviewer 1 for the extensive and careful work that was done for improving the quality of our manuscript.

Reviewer 2 Report
This is a mostly well-written paper describing the discovery of differentially-methylated markers in two major breast cancer subtypes, triple-negative (TNBC) and luminal-B, that are predictive of response to treatment with neoadjuvant chemotherapy according to standard-of-care (at the time of discovery – this is a rapidly evolving area in oncology). The markers were discovered by RRBS in a discovery series of pre-treated, histologically classified, tumor biopsies in adequately-sized discovery series. These were then tested in an independent series of pre-treated tumors of the respective subtypes by MSRE-qPCR.
Overall, this is an interesting study and innovative in being among the first to demonstrate the utility of DNA methylation markers for stratification of tumors for prediction of response to treatment, with relatively high accuracy. Particular clinicopathologic features added value when combined with the methylation marker panel.
The paper would be improved with the following minor inclusions/amendments:
The DNA methylation-based wet-lab methods are fairly standard and custom designs of the MSRE-qPCR assays are adequately described. However, the algorithm/statistics used to select the candidate panels from among the numerous differentially-methylated loci identified (listed in Supplementary data) are not well described.
For Tables 1, 2, 3 and 4:
Provide range of ages as well as medians.
Lymph node “state” should be “status”
“Additional cohort” should be “independent cohort” throughout. Alternatively, they should be termed “discovery” and “validation” cohorts, as appropriate.
These tables could actually be combined into two tables or a single table with “discovery” left, “validation/independent” right, and BC subtypes top and bottom. Features should be compared between the discovery and validation cohorts by each subtype to determine if there are any significant differences, in which case, the P values should also be provided. The ages seem to be older in the validation cohorts than the respective discovery cohorts, which should be made clear.
Line 467-8 “It is found in the endothelium of tumor cells and is a tumor endothelial marker” – omit words “cells”?
Author Response
Point 1: The DNA methylation-based wet-lab methods are fairly standard and custom designs of the MSRE-qPCR assays are adequately described. However, the algorithm/statistics used to select the candidate panels from among the numerous differentially-methylated loci identified (listed in Supplementary data) are not well described.
Response 1: Thank you very much for this comment. We have added a more complete description in the “DNA methylation markers selection criteria” section.
Point 2: For Tables 1, 2, 3 and 4: Provide range of ages as well as medians. Lymph node “state” should be “status”.
Response 2: Thank you. We combined tables for discovery and independent cohorts and provided range of ages (Table 1 and Table 2). The “state” is the “status” throughout the text now.
Point 3: “Additional cohort” should be “independent cohort” throughout. Alternatively, they should be termed “discovery” and “validation” cohorts, as appropriate.
Response 3: Thank you for this comment. We have replaced “additional cohort” by “independent cohort” throughout the document.
Point 4: These tables could actually be combined into two tables or a single table with “discovery” left, “validation/independent” right, and BC subtypes top and bottom. Features should be compared between the discovery and validation cohorts by each subtype to determine if there are any significant differences, in which case, the P values should also be provided. The ages seem to be older in the validation cohorts than the respective discovery cohorts, which should be made clear.
Response 4: Thank you very much. Merging tables is a great solution. Statistical comparison of clinical features between the discovery and independent cohorts with P values were added in “Materials and Methods” section.
Point 5: Line 467-8 “It is found in the endothelium of tumor cells and is a tumor endothelial marker” – omit words “cells”?
Response 5: Thank you. We replaced “the endothelium of tumor cells” by “tumor endothelium”.
We thank Reviewer 2 for the extensive and careful work that was done for improving the quality of our manuscript.

Round 2
Reviewer 1 Report
Revised version of manuscript doesn’t present enough confidence on the significance of identified epigenetic markers. Authors’ major argument is not either clear nor supported by their results. Therefore, the manuscript is not acceptable.
The epigenetic maker combination should be transformed first into countable drug-response factor with the test set of cases, and the transformed countable factor should be employed to classify high risk and low risk groups for drug-responsiveness. Then, with the countable factor, independent set of cases should be classified into high risk and low risk groups, which can be evaluated with their correlation with drug-responsiveness in the independent cases. The P value should be presented after adjusting T and N stages (or age) all of which are significant factors for patients’ survival in independent set of cases.
Author Response
Thank you for for this detailed comment. To investigate the potential predictive value of combined epigenetic and clinical panels, we calculated countable NACT response score for epigenetic panels developed within this study and combined it with independent clinical features, such as tumor size, lymph node in-volvement, clinical stage, and patient’s age. All corrections in the text are marked in yellow.